# IGFs Potentiate TAC3-induced SLα Expression via Upregulation of TACR3 Expression in Grass Carp Pituitary Cells

**DOI:** 10.3390/cells8080887

**Published:** 2019-08-13

**Authors:** Guangfu Hu, Mulan He, Wendy K. W. Ko, Cheng Ye, Qiongyao Hu, Anderson O. L. Wong

**Affiliations:** 1College of Fisheries, Huazhong Agricultural University, Wuhan 430070, China; 2School of Biological Sciences, University of Hong Kong, Hong Kong, China

**Keywords:** signal transduction, tachykinin receptor, pharmacological test, neurokinin B, grass carp

## Abstract

In mammals, the tachykinin 3 (TAC3)/tachykinin receptor 3 (TACR3) systems have been confirmed to play an important role in the regulation of puberty onset. Using grass carp pituitary cells as the model, our recent study found that the TAC3 gene products could significantly induce somatolactin α (SLα) synthesis and secretion via TACR3 activation. In the present study, we seek to examine if pituitary TACR3 can serve as a regulatory target and contribute to TAC3 interactions with other SLα regulators. Firstly, grass carp TACR3 was cloned and tissue distribution showed that it could be highly detected in grass carp pituitary. Using HEK293 cells as the model, functional expression also revealed that grass carp TACR3 exhibited ligand binding selectivity and post-receptor signaling highly comparable to its mammalian counterpart. Using grass carp pituitary cells as the model, TACR3 mRNA expression could be stimulated by insulin-like growth factor (IGF)-I and -II via the IGF-I receptor coupled to phosphatidylinositol-3-kinase (PI_3_K)/protein kinase B (Akt) and mitogen-activated protein kinase (MAPK) pathways. Interestingly, IGF-I/-II cotreatment could also significantly enhance TAC3-induced SLα mRNA expression and the potentiating effect was dependent on TACR3 expression and activation of adenylate cyclase (AC)/cAMP/protein kinase A (PKA), phospholipase C (PLC)/inositol 1,4,5-triphosphate (IP_3_)/protein kinase C (PKC), and Ca^2+^/calmodulin (CaM)/calmodulin-dependent protein kinase II (CaMK-II) cascades. Besides, IGF-I-induced Akt phosphorylation but not MEK, extracellular signal-regulated kinase (ERK_1/2_), and P_38_MAPK phosphorylation was notably enhanced by TACR3 activation. These results, as a whole, suggest that the potentiating effect of IGFs on TAC3 gene products-induced SLα mRNA expression was mediated by TACR3 upregulation and functional crosstalk of post-receptor signaling in the pituitary.

## 1. Introduction

In mammals, tachykinin receptor 3 (TACR3) can preferentially bind with neurokinin B (NKB) but to a lower extent for other tachykinins, and its activation is responsible for a wide range of biological actions including adenylate cyclase (AC)/cAMP/protein kinase A (PKA), phospholipase C (PLC)/inositol 1,4,5-triphosphate (IP_3_)/protein kinase C (PKC), and Ca^2+^/calmodulin (CaM)/calmodulin-dependent protein kinase II (CaMK-II) cascades [1]. The role of TACR3 activation in reproductive functions has aroused a lot of interest in the reproductive biology field mainly due to the recent demonstrations that loss-of-function mutations in TACR3 or its ligand NKB can lead to hypogonadotropic hypogonadism and infertility in human [2,3,4]. The study of the TAC3/TACR3 system has been recently extended to fish models, e.g., zebrafish [5,6,7] and grass carp [8]. Interestingly, unlike the situation in mammals, the TAC3 gene in fish encodes not only the mature peptide of NKB but also the amino acid (a.a.) sequence of a novel tachykinin-like peptide named NKB-related peptide (NKBRP). Besides, two TAC3 genes, with TAC3a coding for NKBa and NKBRPa and with TAC3b coding for NKBb and NKBRPb, have been identified in fish species and are suspected to be the result of 3R whole genome duplication that occurred during the evolution of modern-day bony fish [9]. In our recent studies, the TAC3a gene products, namely NKBa and NKBRPa, were found to stimulate somatolactin α (SLα) secretion and gene expression in grass carp pituitary cells via TACR3 activation and subsequent stimulation of the PLC/IP_3_/PKC, cAMP/PKA, and Ca^2+^/CaM/CaMK-II pathways [8]. Although TACR3 is known to be important in reproductive functions in mammals, not much is known regarding the neuroendocrine regulation of TACR3 expression at the pituitary level. Except for a single report in zebrafish suggesting that brain TACR3 mRNA expression can be modified by estrogen treatment [5], to our knowledge, little is known about TACR3 regulation in fish models. 

Insulin-like growth factors (IGF), including IGF-I and IGF-II, are polypeptides, sharing high levels of sequence homology with insulin [10]. They are produced mainly in the liver as endocrine hormones but also secreted locally at the tissue level as autocrine/paracrine functions and can exert their biological functions by binding to IGF-I receptors (IGF-IRs), which are single-pass membrane receptors with tyrosine kinase activity and functionally coupled with the mitogen-activated protein kinase (MAPK) and phosphatidylinositol-3-kinase (PI_3_K)/protein kinase B (Akt) signal pathways [11]. At the pituitary level, e.g., in human and rodents, IGF-I is well-documented to act as a negative feedback regulator for growth hormone (GH) secretion and gene expression [12]. Similarly, GH could also induce IGF-I expression in fish liver [13] and IGF could also inhibit GH release and gene expression in pituitary cells prepared from eel [14] and grass carp [15]. In fish models, somatolactins including SLα and SLβ, are the latest members identified for the GH gene lineage [16,17] and are involved in important biological functions including background adaptation, acid–base balance, reproduction, and immune cell activation [18]. Unlike the inhibitory action of IGF on GH release via long-loop feedback, our recent studies in grass carp pituitary cells have shown that IGF-I/-II could serve as potent stimulators for SLα and SLβ secretion and gene expression via the MAPK-dependent pathway coupled to IGF-IR expressed at the pituitary level [19].

In mammals, function crosstalk between insulin/IGF-I and G-protein coupled receptor (especially G_q_-coupled GPCR) have been well-documented and believed to play a role in cancer progression, e.g., in pancreatic cancer [20]. In the case of ductal pancreatic adenocarcinoma, insulin/IGF-I receptor activation is known to exhibit function crosstalk with neurotensin and AT1 receptor post-receptor signaling via the G_q_/PKC/PKD/MEK and PI_3_K/Akt/mTOR/[Ca^2+^]_i_ pathways, which may lead to the potentiation of DNA synthesis and cancer cell proliferation [21,22]. In rabbit, the synergistic effect of Substance P (SP) and IGF-I cotreatment has been reported in the process of corneal epithelial wound closure [23], which can be attributed to the enhancement of epithelial cell migration on the cornea by increasing their physical association with extracellular matrix protein [24] via TACR1 activation coupled to the PLC/IP_3_ and Ca^2+^/CaMK-II pathways [25]. Recently, our lab found that cotreatment with somatolactins, IGFs, and TAC3 gene product together could serve as a highly potent stimulatory signal for SLα mRNA expression. A further study found that somatolactin could enhance the TAC3 gene product-induced SLα mRNA expression via upregulation of TACR3 expression [26]. However, the reason why IGF could further enhance the potentiating effects of somatolactin on TAC3 gene product-induced pituitary SLα mRNA expression was not clear.

In China, grass carp (*Ctenopharyngodon idellus*) is the most important aquaculture species, of which the total production has reached 5.50 million tonnes in 2018. Generally, grass carp need to take about five years to reach sexual maturity, which severely restricts the selection of breeding processes. Considered high market demands and long-term growth periods of grass carp, revealing the regulatory mechanisms in sexual maturation of grass carp will be crucial for grass carp breeding. In mammals, previous studies have demonstrated that the TAC3/TACR3 system played pivotal roles in the regulation of reproduction and puberty onset [3]. Our recent studies have also found that the TAC3/TACR3 system should also play an important role in grass carp puberty onset via regulation of pituitary SLα secretion and synthesis [8]. Now, we would like to know whether other factors could enhance TAC3/TACR3 system-regulated puberty onset in grass carp. 

In the present study, grass carp TACR3 has been cloned, and its tissue expression profile, especially its expression at the pituitary level, has been established. Using human HEK-293 cells with stable expression of grass carp TACR3, the ligand selectivity and functional coupling with post-receptor signaling of this newly cloned receptor were also characterized. Using grass carp pituitary cells as the model, we also have novel findings that TACR3 mRNA expression could be upregulated by IGF-IR activation which could be correlated with a synergistic effect of IGF-IR and TACR3 activation on SLα gene expression. Using a combination of pharmacological approaches and direct probing of various signaling kinase, the post-receptor signaling mechanisms responsible for TACR3 upregulation and the synergistic effect on SLα gene expression were also examined. Our study for the first time provides evidences that IGF-I/-II could significantly enhance TAC3 gene product-induced pituitary hormone gene expression via upregulation of TACR3 gene expression in grass carp. The present study also suggested that IGFs might enhance TAC3-regulated puberty onset in teleost. 

## 2. Materials and Methods

### 2.1. Animals and Reagents

One-year-old (1+) grass carps (*Ctenopharyngodon idellus*) with body weights ranging from 1.5–2.0 kg were acquired from local markets and maintained in 250-L aquaria under 12D:12L photoperiod at 20 °C. Since the grass carp at this stage was prepubertal and sexual dimorphism was not apparent, fish of mixed sexes were used for pituitary cell preparation according to the protocol approved by the committee for animal use at Huazhong Agricultural University (Ethical Approval No. HBAC20091138; Date: 15 November 2009). Grass carp NKBa, NKBRPa, NKBb, NKBRPb, SP, NKA, and FAM-NKBa synthesized by GenScript Corporation (Nanjing, China) were dissolved in DMSO and stored frozen at −80 °C as 1 mM stocks in small aliquots. Human IGF-I, IGF-II, and insulin were purchased from Sigma (St. Louis, MO, USA), dissolved in double-distilled deionized water, and stored as 0.1 mM stocks in small aliquots at −80 °C. The other drugs for receptor specificity and post-receptor signal pathway were listed in Appendix A. In our Western blot studies, the antibodies used for the detection of total and phosphorylated extracellular signal-regulated kinase (ERK_1/2_) were purchased from Sigma, whereas the antibodies for total and phosphorylated MEK_1/2_, total and phosphorylated Akt, and phosphorylated and total p_38_MAPK were purchased from Cell Signaling (Danvers, MA, USA) (Appendix A). The secondary antibodies (raised in mouse or rabbit) conjugated with horseradish peroxidase were purchased from KPL (Gaitherbug, MD, USA). 

### 2.2. Molecular Cloning, Tissue Expression, and Structural Analysis of Grass Carp TACR3

Total RNA prepared from grass carp pituitary was reversely transcribed using the Superscript II First Strand cDNA Synthesis Kit from Invitrogen (Carlsbad, CA, USA). Based on the sequence alignment of TACR3 cDNAs reported in other fish species, primers were designed to amplify the conserved region of TACR3 transcripts in grass carp. Full-length cDNA obtained using 5′/3′ RACE for grass carp TACR3 was then compiled using the MacVector 9.5.2 software (Oxford Molecular, Madison, WI, USA). Phylogenetic analysis based on TACR3 nucleotide sequences was conducted with MEGA 6.0 using the neighbor-joining method [27]. The 3-D protein models of TACR3 were constructed with the SWISS-MODEL server [28]. For tissue distribution of TACR3 in grass carp, RT-PCR was conducted in RNA isolated from various tissues and selected brain areas using primers specific for grass carp TACR3 (Appendix A). In these studies, RT-PCR of β-actin was performed to serve as an internal control. 

### 2.3. Functional Expression of Grass Carp TACR3 in HEK-293 Cells

To clarify the ligand selectivity and signal transduction pathways of TACR3, functional expression of grass carp TACR3 was performed in HEK-293 cells with co-transfection of luciferase-expressing reporters targeting different signaling pathways provided by the Clontech Cell Signaling Profiling Kit (Clontech, Mountain View, CA, USA), including cAMP response element (CRE)-Luc (for cAMP pathway), activator protein-1 (AP1)-Luc (for PKC pathway), nuclear factor of activated T-cells (NFAT)-Luc (for Ca^2+^/calcineurin pathway), and serum responsive element (SRE)-Luc (for MAPK pathway) [29]. The ORF of grass carp TACR3 (GenBank no: JQ254913) was ligated with the eukaryotic expression vector pcDNA3.1/Zeo(−) (Invitrogen, Carlsbad, CA, USA) to generate the TACR3 (pcDNA3.1/Zeo(−)-TACR3) expression vector. For transfection experiments, HEK-293 cells with stable expression of grass carp TACR3 were seeded at a density of 0.05 × 10^6^ cells/0.5 mL/well in 24-well plates. After overnight incubation for recovery, transfection was carried out in OPTI-MEM (Gibico, Carlsbad, CA, USA) with a luciferase-expressing reporter for the respective pathway, pTK.RL (Promega, Madison, WI, USA), pEGFP-N1 (Clontech), and lipofectamine 2000 (Invitrogen). pTK.RL, the renilla luciferase-expressing reporter, was included to serve as an internal control, whereas the GFP-expressing vector pEGFP-N1 was used to monitor the transfection efficiency between individual wells and separated experiments. After transfection, the cells were allowed to incubate for 18–24 h at 37 °C in DMEM supplemented with 10% FBS before drug treatment. After 24 h drug treatment, the cells were washed with ice-cold PBS and dissolved in passive lysis buffer (Promega). Lysate samples of the cells prepared were assayed for both firefly luciferase and renilla luciferase activities using a Dual-Glo^TM^ luciferase Assay Kit (Promega) in a Lumat LB9507 Luminometer (EG&G, Gaithburg, MD, USA). Transfection experiments were performed in quadruplicate with cells cultured in separate wells.

To further clarify the functional role of TACR3 on [Ca^2+^]_e_ entry and [Ca^2+^]_i_ mobilization, clonal expression of HEK-293 cells with stable expression of grass carp TACR3 was prepared from a single colony of HEK-293 cells transfected with the pcDNA3.1/Zeo(−) expression vector for grass carp TACR3, followed by 3-month selection with Zeocin (Invitrogen). The Ca^2+^ responses in these TACR3-expressing cells were monitored using a Ca^2+^ bioluminescence plate assay as described previously [30]. Briefly, the aequorin-expressing plasmid mtAEQ/pcDNA1 (Molecular Probe; Eugene, OR) was transient transfected into HEK-293 cells with stable expression of grass carp TACR3. After 12 h recovery in DMEM with 10% FBS, the cells were detached by trypsin (0.5 mg/mL) treatment and transferred to 96-well microtitre white plate (Waltham, MA, USA) at a density of 40,000 cells/100 μL/well and incubated for another 24 h for cell attachment. To constitute the holoenzyme apoeaequorin for Ca^2+^ signal detection, HEK-293 cells were incubated with 90 μL/well calcium-free DMEM medium (for probing intracellular Ca^2+^ mobilization) or DMEM (for probing total Ca^2+^ signals) containing 5 μM coelenterazine (Invitrogen) for 3 h in the dark at 37 °C under 5% CO_2_ and saturated humidity. After that, the cells were challenged with different concentrations of test substances administered in a volume of 10 μL in PBS. Luminescence signals indicative of a rise in intracellular Ca^2+^ levels were monitored using a FluoStar OPTIMA (BMG Labtech, Ortenberg, Germany) plate reader at room temperature. For dynamic measurement of luminescence signal, light emission was recorded every 0.5 s over a period of 50 s for individual wells. The total luminescence response after drug treatment was defined as the sum of the net changes of luminescence signals (i.e., area under the curve) over a period of 50 s after the initiation of drug treatment. 

### 2.4. Receptor Binding Assay for Grass Carp TACR3

To evaluate the receptor binding of grass carp TACR3 with tachykinin peptides, HEK-293 cells with stable expression of grass carp TACR3 were grown to 60–70% confluence in T75 flasks. The cells were then harvested and seeded onto 24-well plates at a density of 0.1 million/well and incubated at 37 °C under 5% CO_2_ for 24 h prior to the experiments. On the day of the experiments, the culture medium was aspirated and replaced with 500 μL serum-free DMEM with or without appropriated levels of unlabeled competitors. Cells were then allowed to incubate for 30 min at 37 °C under 5% CO_2_ and saturated humidity. FAM Carboxyfluorescein-labeled NKBa, namely FAM-NKBa, was used as a tracer for receptor binding experiments, and a 50-μL volume of FAM-NKBa (10 nM) was added into individual wells. After further incubation for 30 min at 37 °C, the cells were washed twice and with 500 μl PBS each to remove unbound ligands. After that, the cells were treated with 120 μL ligand dissociation buffer (50 mM Tris-HCl, 150 mM NaCl, 1 mM EDTA, 1% NP-40, and 0.25% Na deoxycholate). The buffer with FAM-NKBa released from TACR3 binding was harvested after the cell debris was removed by high speed centrifugation at 4 °C. The supernatant from individual wells was then collected in a 96-wells black view plate, and fluorescence signals indicative of FAM-NKBa binding were registered using FluoStar OPTIMA with excitation at 490 nm and emission at 520 nm.

### 2.5. Fluorescence Imaging of TACR3 Binding

To evaluate the receptor binding of TACR3 visually, HEK-293 cells were transiently transfected with the blank vector pcDNA3.1 or grass carp TACR3-expressing vector pcDNA3.1/zeo-TACR3 and cultured overnight at a density of 0.5 million cells/mL on poly-D-lysine coated coverslip. On the following day, the old medium was removed and the cells were incubated with 1 μM FAM-NKBa for 30 min at 37 °C under 5% CO_2_ and saturated humidity. The cells were rinsed three times with 1 × PBS and fixed with 4% paraformaldehyde at room temperature for 15 min. The coverslip with cells attached was mounted onto glass slide and viewed under a LSM 710 confocal microscopy (Carl Zeiss, Oberkochen, Germany). The cells with successful transfection were identified by mCherry fluorescence detected with excitation at 587 nm and emission at 610 nm. Specific signals for FAM-NKBa binding in transfected cells were registered with excitation at 490 nm and emission at 520 nm.

### 2.6. RT-PCR for TACR3 and IGF-IR Expression in Immune-Identified Somatolactin (SL) Cells

To confirm the co-expression of TACR3 and IGF-IR in SLα-producing cells, the neurointermediate lobe (NIL) with high level of SL cells was manually dissected from twelve grass carp pituitaries under a stereomicroscope and dispersed using trypsin/DNase digestion [31]. The NIL cells were spread evenly onto glass slides (~5 × 10^4^ cells/slide), fixed in Bouin’s fixative, and subjected to immunostaining with antisera for grass carp SLα (1:100,000) and SLβ (1:100,000) using a Vectastain ABC Kit (Vector Lab, Burlingame, CA, USA) [8]. After that, immune-identified SLα cells and SLβ cells (~250 cells per sample) were isolated separately by laser capture microdissection (LCM) using a PixCell-II Cell Isolation System (Arcturus, Mountain View, CA, USA) [32]. Total RNA was extracted from individual cell types and reversely transcribed for PCR detection of grass carp TACR3 and IGF-IR as above. In this study, the NIL mix cells were used as the positive control.

### 2.7. Measurement of SLα and TACR3 mRNA Expression

Grass carp pituitary cells prepared by the trypsin/DNase digestion method [31] were seeded in 24-well plates at ~2.5 × 10^6^ cells/mL/well. On the following day, drug treatment was initiated by replacing the old medium with a testing medium with the appropriate levels of test substances, and the cells were then allowed to incubate at 28 °C for the duration as indicated. After drug treatment, total RNA was extracted from individual wells and reversely transcribed as above. The RT samples obtained were subjected to qPCR using a LightCycler SYBR Green I Kit (Roche, Stockholm, Sweden) with the primers mentioned above. In these studies, serial dilutions of plasmid DNA containing the ORF of SLα (GeneBank no.: EF372074) and TACR3 (GenBank no: JQ254913) cDNA were used as the standards for data calibration. Parallel real-time PCR measurement of β-actin was also conducted in individual experiments to serve as the internal control. 

### 2.8. Western Blot for Signaling Kinases

The pituitary cells derived from twenty grass carp pituitaries were seeded in 24-well culture plates at a density of 2.5 × 10^6^ cells/mL/well and incubated with drug treatment for the duration as indicated in the individual experiments. For detection of various kinase protein phosphorylation, the duration of drug treatment was reduced to 30 min based on time course validation. After drug treatment, the culture medium from individual wells was removed and the remaining cells were rinsed with PBS and lysed in RIPA buffer (50 mM Tris^·^HCl, 150 mM NaCl, 1 mM EDTA, 1% NP-40, and 0.25% Na deoxycholate), containing a final concentration of 1×protease/phosphatase inhibitor cocktail (Roche). The cell lysate was cleared by high-speed centrifugation at 4 °C, the clear supernatant was resolved in 10% gel by SDS-PAGE, and Western blotting was conducted according to the procedures as previously described [19]. The antibodies that can be detected, the phosphorylated and total ERK_1/2_ (1:5,000), MEK_1/2_ (1:1,000), p_38_ MAPK (1:1,000), and Akt (1:1,000), were used at the dilutions recommended by the manufactures. Following the incubation, the membranes were washed three times by TBS-T to remove nonspecific binding of primary antibodies and the HRP-conjugated secondary antibodies (goat anti-rabbit IgG (1:5000)) were introduced for signal development. Chemiluminescence signals for target immune-reactivity were detected using SuperSignal West Pico (PIERCE, Rockford, IL, USA) as the substrate and quantified using the IC440 CF Digital Science Image Station (Eastman Kodak, Rochester, NY, USA). In these experiments, Western blot of β-actin was used as an internal control by using its antibody (1:15,000; Oncogen, Cambrige, MA, USA). 

### 2.9. Data Transformation and Statistics

For real-time PCR of SLα and TACR3 mRNA measurement, standard curves with a dynamic range of ≥10^5^ and correlation coefficient of ≥0.95 were used for data calibration with the RotorGene-Q software 1.7 (Qiagen) under unsupervised mode. SLα and TACR3 mRNA data were normalized with the β-actin transcript level and then were transformed as a percentage of the mean value in the control group without drug treatment (as “%Ctrl”). For the calcium bioluminescence signals and raw data obtained from luciferase reporter assays, the luminescence signals or luciferase activity was expressed as “relative luminescence unit (RLU)” registered by the fluorescence plate reader. For individual samples of the luciferase reporter assay, the firefly luciferase activity detected was routinely normalized against Renilla Luciferase activity expressed in the same well and expressed as the luciferase activity ration (as “Luc Ratio”). For receptor binding assay, specific binding of FAM-NKBa was measured in terms of “Arbitrary Fluorescence Unit (AFU)”. The six replicates presented (expressed as Mean ± SEM) were pooled results from two individual experiments prior to statistical analysis; all data were tested for normality of distribution using the Shapiro–Wilk normality test. Then, one-way ANOVA and two-way ANOVA were used to test the significant difference according to different experiment groups. The differences between groups were considered as significant at *P* < 0.05 (“*”) or highly significant at *P* < 0.01 (“**”).

## 3. Results

### 3.1. Molecular Cloning and Tissue Distribution of Grass Carp TACR3

The full-length TACR3 cDNA (GenBank no.: JQ254913) was isolated from the grass carp pituitary (Appendix A). The a.a. sequence of grass carp TACR3 is only 63.8–64.8% homologous to that of the mammalian counterparts, but the regions for transmembrane domain TMD1–3 and 6–7 as well as the intracellular loops ICL1–2 are highly conserved among vertebrate species (Figure 1A and Appendix A). In silico protein modeling also confirms that the 3-D structures of the transmembrane domains are highly comparable with that of its human counterpart (Figure 1C). Phylogenetic analysis based on the nucleotide sequence obtained further confirms that the newly cloned receptor is an orthology of vertebrate TACR3 and can be clustered in the clade of TACR3 subfamily with a close evolutionary relationship with zebrafish TACR3 (Figure 1B). RT-PCR also revealed that, except for the spleen, kidney, muscle, and various brain areas, TACR3 gene could be detected at high levels in the heart, intestine, liver, and pituitary and to a lower extent in the gill and gonads (Figure 1D).

### 3.2. Functional Expression of Grass Carp TACR3

As a first step in evaluating the ligand selectivity of the newly cloned receptor, a line of human HEK-293 cells with stable expression of grass carp TACR3 was established and used for a transfection study with a luciferase-expressing CRE-Luc reporter and receptor binding assays with various TAC3 gene products and subtype-specific NKR agonists. In this case, luciferase activity expression mediated by CRE sites with the promoter of the CRE-Luc reporter could be increased in a dose-dependent manner by treatment with the TACR3 specific agonist senktide (with ED_50_ at 21.3 ± 3.2 nM) but with a lower potency and efficacy for the TACR2-specific agonist GR64349 (with ED_50_ at 2161 ± 40.8 nM) and the TACR1-specific agonist Hemokinin-1 (with ED_50_ at 4096 ± 113.2 nM) (Figure 2A). Parallel activity studies using grass carp TAC1 and TAC3 gene products also revealed that grass carp NKBa, NKBRPa, and NKBRPb were found to be notably more potent in triggering the corresponding stimulatory effects (with minimal effect doses in nanomolar dose range) compared to SP, NKA, and NKBb (with minimal effect doses in micromolar dose range) (Figure 2B). Besides, HEK-293 cells with stable expression of grass carp TACR3 were also transiently transfected with SRE-Luc, AP1-Luc, NFAT-Luc, or mtAEQ/pcDNA1 luciferase-expressing vectors and to allow for functional evaluation of the activation status of PKC- and Ca^2+^-dependent pathways, respectively. As shown in Table 1, grass carp NKBa (ED_50_: 15.5–50.3 nM), NKBRPa (ED_50_: 17.9–38.6 nM), and NKBRPb (ED_50_: 12.7–40.4 nM) were also found to be the most potent in triggering the corresponding stimulatory effects (with minimal effect dose in nanomolar dose range) compared to SP (ED_50_: 487.1–1071.3 nM), NKA (ED_50_: 392.3–1512.3 nM), and NKBb (ED_50_ > 10 μM). To further characterize the receptor binding affinity and selectivity of grass carp TACR3, a fluorescence-based receptor binding assay using the FAM-tagged NKBa as a tracer was also conducted in HEK-293 cells with stable expression of grass carp TACR3. As shown in Figure 2C, the TACR3 agonist senktide was effective in displacing FAM-NKBa binding in the nanomolar dose range (with IC_50_ at 71.2 ± 9.23 nM) whereas noticeable displacement of the fluorescence signal could be observed only at the micromole level for the TACR1 agonist hemokinin-1 (with IC_50_ > 10 μM) and TACR2 agonist GR64349 (with IC_50_ >10 μM). Parallel displacement studies using grass carp TAC1 and TAC3 gene products also revealed that the specific binding of grass carp TACR3 could be displaced in the nanomolar dose range with the TAC3 gene products including NKBa (with IC_50_ at 17.94 ± 1.41 nM), NKBRPa (with IC_50_ at 13.99 ± 2.08 nM), and NKBRPb (with IC_50_ at 16.53 ± 1.52 nM) and to a lower micromolar dose range for TAC1 gene products including SP (with IC_50_ at 926.5 ± 56.15 nM) and NKA (with IC_50_ at 953.0 ± 30.43 nM). Interestingly, the other TAC3 gene products, namely NKBb, was found to be the least effective in displacing the specific binding of FAM-NKBa on grass carp TACR3 (with IC_50_ > 10 μM) (Figure 2D). To further evaluate the receptor binding of TACR3 visually, confocal fluorescence imaging was performed with FAM-NKBa as a fluorescence-tagged ligand in HEK-293 cells after transient transfection with the grass carp TACR3 expression vector pcDNA3.1-Zeo-TACR3 (Figure 2E). In our validation study, fluorescence signal of FAM-NKBa was detectable mainly in the periphery of transfected cells with mCherry expression after a brief incubation (30 min) of HEK-293 cells with FAM-NKBa following transient transfection of the grass carp TACR3 expression vector (Figure 2E). However, the FAM-NKBa fluorescence signals were not found in untransfected cells (without mCherry expression) or in parallel experiments with transfection of a blank vector without the TACR3 insert (Figure 2E).

### 3.3. IGF-Induced TACR3 mRNA Expression in Grass Carp Pituitary Cells

As a first step in establishing the anatomic basis for the functional interaction between TAC3 gene products and IGFs, RT-PCR for IGF-IR and TACR3 expression were conducted in immuno-identified grass carp SLα and SLβ cells isolated by the laser capture microdissection (LCM) technique (Figure 3A). In this case, IGF-IR signals were located in the SLα and SLβ cells, while the PCR signals for grass carp TACR3 were only found in SLα cells but not in SLβ cells (Figure 3A). To examine the direct effect of IGF-IR activation on TACR3 expression at the pituitary level, the primary culture of grass carp pituitary cells was challenged with human IGF-I and IGF-II. Our previous study has confirmed that grass carp IGFs and IGF-IR were highly conserved with that in mammals and that mammalian IGFs were known to activate IGF-IR in fish models [13], so human IGF-I/-II were used as the stimulant for IGF-IR in the present study. The time-course experiment revealed that IGF-I/-II could significantly stimulate TACR3 mRNA expression from 1 h to 24 h in a time-dependent manner (Figure 3B). In the parallel dose-dependent studies, a 24-h incubation with increasing levels of IGF-I or IGF-II (0.01–100 nM) could also trigger TACR3 mRNA expression in a concentration-related fashion; however, this stimulatory effect was not mimicked by similar treatment with increasing levels of insulin (Figure 3C). To further clarify the receptor specificity for TACR3 regulation by IGFs, a pharmacological approach was used. As shown in Figure 3D, the stimulatory effects of IGF-I on TACR3 mRNA expression was totally abolished/partially inhibited by simultaneous treatment with the IGF-IR inhibitors PQ401 and PPP, respectively, while the inhibitor for insulin receptor, HNMPA, was not effective in this regard. With the use of pharmacological blockers targeting different signaling pathways, the signal transduction mechanisms for TACR3 expression regulated by IGF-IR activation was examined. As shown in Figure 3E, the stimulatory effects on TACR3 mRNA expression induced by IGF-I treatment were notably suppressed by cotreatment with the wortmannin (PI_3_K inhibitor), HIMOC (Akt inhibitor), and rapamycin (mTOR inhibitor). Furthermore, cotreatment with the PD98059 (MEK_1/2_ inhibitor), SB203580 (p_38_MAPK inhibitor), or JNK2 inhibitors could also inhibit the stimulatory effect of IGF-I on TACR3 mRNA expression in grass carp pituitary cells (Figure 3F).

#### Synergistic Effects of IGF and TAC3 Gene Products on SLα mRNA Expression

According to our previous studies, both IGF-IR and TACR3 were detected in SLα cells (Figure 3A) and IGF-I/-II treatment could also stimulate SLα synthesis and secretion in grass carp pituitary cells [19]. To further examine the functional role of IGF-induced TACR3 gene expression on SLα expression, cotreatment of IGF-I/-II with TAC3 gene products was performed in the grass carp pituitary cells. As shown in Figure 4A, IGF-I and NKBRPa alone could both induce SLα mRNA levels in grass carp pituitary cells in a time-dependent manner. Interestingly, IGF-I could significantly enhance the NKBRPa-induced SLα transcript level (up to 10 folds basal), especially after 12–24 h of drug treatment. In addition, the potentiating effect could still be examined with cotreatment of IGF-I/-II with the grass carp TAC3 gene products including NKBRPa, NKBa, NKBRPb, and NKBb (Figure 4B). In parallel studies, the dose-dependence of this synergistic action was also confirmed. In this case, NKBRPa-stimulated SLα mRNA expression was enhanced in a concentration-related fashion with simultaneous treatment with increasing levels of IGF-I (0.01–100 nM; Figure 4C). Similarly, in the reciprocal experiment, IGF-I-induced SLα mRNA expression could also be dose-dependently enhanced by increasing concentrations of NKBRPa (0.1–1000 nM; Figure 4D). To build the functional interaction between IGF potentiation and TACR3 expression at the pituitary level, the TAC3 gene products were replaced with TACR3 agonist senktide in the potentiation study with IGF cotreatment. In this case, TACR3 activation with senktide was found to mimic the synergistic effects of TAC3 gene products on SLα mRNA expression when given together with either IGF-I or IGF-II (Figure 4E). Furthermore, cotreatment with SB222200 (TACR3 antagonist) not only could suppress IGF-I- or NKBRPa-induced SLα transcript expression but also markedly reduced the potentiating effect induced by IGF-I + NKBRPa (Figure 4F). These results indicate that the potentiating effect of IGF and TAC3 gene products on SLα mRNA expression is dependent on TACR3 activation. 

### 3.4. Signal Transduction for the Synergistic Effect of IGFs and TACR3 Activation

To clarify the post-receptor signaling mechanisms involved in the potentiating effect of IGFs and TAC3 gene products on SLα mRNA expression, a pharmacological approach was used. In this case, pituitary cells were exposed to IGF-I and senktide alone or in combination with/without cotreatment with various inhibitors targeting the PLC/IP_3_/PKC, cAMP/PKA, and Ca^2+^/CaM/CaMK-II pathways (Figure 5). In these experiments, simultaneous incubation with U73122 (PLC inhibitor) (Figure 5A), GF109203X (PKC inhibitor) (Figure 5A), 2-APB (IP3 receptor blocker) (Figure 5A), MDL12330A (AC inhibitor) (Figure 5B), H89 (PKA inactivator) (Figure 5B), calmidazolium (CaM antagonist) (Figure 5C), nifedipine (VSCC blocker) (Figure 5C), and KN62 (CaMK-II inactivator) (Figure 5C) not only could attenuated the stimulatory action on SLα mRNA expression induced by IGF-I or senktide treatment alone but also notably suppressed the synergistic effect caused by IGF-I and senktide cotreatment in grass carp pituitary cells. To further evaluate if IGF-I cotreatment can potentiate the stimulatory effects on SLα gene expression mediated through the cAMP-, PKC-, and Ca^2+^-dependent cascades, pituitary cells were also challenged with IGF-I in the presence of various stimulators/activators for the respective signaling cascades. In these cases, static incubation for 24 h with the AC activator forskolin (Figure 5D), membrane-permeant cAMP analog 8Br.cAMP (Figure 5D), DAG analog Dic8 (Figure 5D), PKC activator TPA (Figure 5D), Ca^2+^ ionophore A23187 (Figure 5D), and VSCC activator BayK8644 (Figure 5D) were all effective in elevating SLα mRNA levels in grass carp pituitary cells. However, cotreatment with IGF-I in the respective studies either induced an additive effect (for the activators for cAMP- and Ca^2+^-dependent cascades) or did not further enhance the stimulation on SLα transcript expression (for the activators for PKC-dependent pathway). In parallel experiments, the effects of TACR3 activation on the signal transduction pathways coupled to IGF-IR activation, namely MAPK and PI_3_K/Akt cascades, were also examined using Western blot to probe the phosphorylation status of the respective signaling kinase. In this case, senktide cotreatment has a noticeable potentiating effect on IGF-I-induced protein phosphorylation of Akt (Figure 6A) but not p_38_MAPK (Figure 6B), MEK_1/2_ (Figure 6C) or ERK_1/2_ (Figure 6D).

## 4. Discussion

After the initial findings that loss-of-function mutations in TAC3 and TACR3 can lead to familial hypogonadotropic hypogonadism and infertility in human and that impairment of the TAC3/TACR3 system can postpone puberty in animal models, increasing evidence has been cumulated supporting the idea of GnRH and kisspeptin regulation by the TAC3/TACR3 system [33,34,35]. However, little information is available regarding the role of TACR3 expression as a regulatory target, especially in lower vertebrates including fish and amphibians. In the present study, to shed light on the role of TACR3 expression on the pituitary action of TAC3 gene products in fish model, grass carp TACR3 has been cloned and phylogenetic analysis based on its cDNA sequence has confirmed that it is an orthologue of vertebrate TACR3 and closely related to TACR3 identified in zebrafish [5,6]. The deduced protein sequence also reveals that grass carp TACR3 is homologous to its mammalian counterparts, especially in the regions covering TMD1-3, TMD6-7, and ICL1-2. 

Based on the results of RT-PCR, TACR3 was found to be expressed in a tissue-specific manner with high levels in the heart, intestine, liver, and pituitary and to a lower extent in the gills and gonad. In mammals, TACR3 is well-documented to be expressed in the gastrointestinal trait and mediates tachykinin regulation of gut motility and secretory functions [36]. The cardiovascular functions of TACR3 has also been reported in mammalian models, e.g., activation of the NKB/TACR3 system is known to increase the heart rate in the dog [37] and rat [38] and to induce vasoconstriction in hepatic portal vein in the rat [39]. In rodents, TACR3 expression can be located in various areas including the corpora lutea, theca, and granulose cells of the ovary [40], suggesting a possible role in follicular growth/maturation. TACR3 expression is also detected in the testes, especially in mature spermatozoa [41], and recent studies have shown that TACR3 together with TACR1 and TACR2 play an important role in the control of sperm motility induced by local production of tachykinins including SP, NKA, and NKB [42]. In our study, a high level of TACR3 was noted in the pituitary, which is consistent with our recent findings of upregulation of SLα synthesis and secretion via TACR3 activation induced by TAC3 gene products in grass carp pituitary cells [8]. 

In mammals, three different subtypes of tachykinin receptors, namely TACR1, TACR2, and TACR3, have been reported and each of them is known to have its own pattern of ligand selectivity, with TACR1 preferring SP (SP > NKA > NKB), TACR2 preferring NKA (NKA > NKB > SP), and TACR3 preferring NKB (NKB > NKA > SP) [43]. Although fish TACR3 has been cloned recently, e.g., in zebrafish [5], the rank order of ligand selectivity based on native ligands and receptor-specific agonists/antagonists has not been fully characterized. In this study, using grass carp TACR3 expressed in HEK-293 cells transfected with CRE-Luc, AP1-Luc, SRE-Luc, and NFAT-Luc reporters or with Ca^2+^-sensitive aequorin vector for functional testing, we have confirmed that the newly cloned TACR3 could be functionally coupled with PKA-, PKC-, and Ca^2+^-dependent signaling mechanisms. Comparison of the respective ED_50_ for functional activation of TACR3 indicated that the grass carp TACR3 can exhibit a rank order of NKBa ≌ NKBRPa ≌ NKBRPb > SP ≌ NKA >> NKBb for receptor activation of various signal pathways, which is consistent with previous reports that TACR3 prefers NKB in mammals [43]. Furthermore, our results based on the dose-dependent induction of CRE-Luc reporter activity and competitive displacement studies using a fluorescence-based receptor binding assay have revealed that grass carp TACR3 could bind and become activated by the TACR3 agonist senktide in the nanomolar dose range but that similar effects would require micromole levels of the TACR1 agonist hemokinin-1 and TACR2 agonist GR64349. Comparison of the respective IC_50_s for ligand displacement and ED_50_ for functional activation of TACR3 indicate that the grass carp TACR3 can exhibit a rank order of senktide >> GR64349 ≥ hemokinin-1 for ligand binding and receptor activation, which is highly comparable with the previous report on rat TACR3 expressed in COS cells [44]. These findings, as a whole, indicated that (i) the newly cloned TACR3 cDNA indeed encode a functional TACR3 in grass carp and (ii) the pharmacological properties of grass carp TACR3 in terms of ligand selectivity and functional coupling to signals pathways are in agreement with that reported in mammalian models [43].

Although IGF-I is well-documented as a feedback inhibitor in the somatotropic axis in vertebrate species [45] and its inhibitory actions on GH secretion and gene expression have also been demonstrated in fish model, e.g., eel [14] and grass carp [15], it also has regulatory actions on other pituitary hormones in the fish pituitary. For examples, stimulatory effects of IGF-I on prolactin (PRL) secretion and gene expression has been reported in hybrid striped bass [46] while similar stimulation on LH and FSH secretion has also been observed in salmon [47] and European eel [48] pituitary cell cultures. In cyprinid species, somatolactin expression at the pituitary level, including SLα and SLβ, is under the control of hypothalamic factors, including PACAP [49,50] and somatostatin [51], as well as endocrine signals from peripheral tissues, e.g., IGF-I and -II [19]. More recently, a novel autocrine/paracrine component with SLα- and SLβ-induced SLα secretion and gene expression have been demonstrated in grass carp pituitary cells [52], suggesting the presence of somatolactin autoregulation occurring at the pituitary level. In the present study, using RT-PCR coupled with LCM isolation of immuno-identified grass carp SLα and SLβ cells, co-localization of IGF-IR and TACR3 was demonstrated in SLα cells but not in SLβ cells, which provides the anatomical basis for functional interactions between IGFs and TAC3 gene products on SLα regulation. This idea has also been confirmed by our novel findings that IGF-I/-II cotreatment could markedly enhance (up to 8-fold increase) the stimulatory effects of TAC3 gene products on SLα mRNA expression in grass carp pituitary cells. At the pituitary level, this potentiating effect could be correlated with IGF-I/-II induction of TACR3 mRNA expression via direct stimulation in pituitary cells in a time- and dose-dependent manner. Since (i) these stimulatory effects of IGF-I and -II were not mimicked by insulin treatment; (ii) IGF-induced TACR3 gene expression could be blocked/inhibited by the IGF-IR inhibitor (e.g., PQ401 and PPP) but not the insulin inhibitor (e.g., HNPMA); and (iii) TACR3 gene expression induced by IGF-I was significantly suppressed by the pharmacological inhibitors for MEK_1/2_ (e.g., PD98059), p38 MAPK (e.g., SB203580), JNK_1/2_ (e.g., JNKII inhibitor), PI3K (e.g., wortmannin), Akt (e.g., HIMOC), and mTOR (e.g., Rapamycin), it is logical to conclude that IGF-IR activation by IGF-I/-II and subsequent stimulation of MAPK- and PI3K/Akt-dependent pathways could trigger upregulation of TACR3 expression in grass carp pituitary, which presumably can contribute to the potentiating effect of IGF-I/-II on SLα gene expression induced by TAC3 gene products. This hypothesis is also consistent with the observations (i) that TACR3 gene expression induced by IGF-IR activation (first observed at 1 h after IGF-I/-II treatment) occurred prior to the onset of the potentiating effect on SLα mRNA expression caused by IGF-I and NKBRPa cotreatment (first observed only after 6 h with drug treatment); (ii) that cotreated with senktide, a TACR3 agonist, was effective in inducing the potentiating action similar to TAC3 gene products when given together with IGF-I/-II; (iii) the synergistic effect on SLα gene expression induced by NKBRPa and IGF-I cotreatment could be notably suppressed by the TACR3 antagonist SB222200; and (iv) a similar potentiating effect caused by senktide and IGF-I cotreatment was also sensitive to pharmacological blockade of the post-receptor signaling cascades previously shown to be coupled with TACR3-mediated SLα gene expression [8], including AC/cAMP/PKA blockade using the AC inhibitor MDL12330A, PKA inhibitor H89, PLC/IP_3_/PKC blockade using the PLC inhibitor U73122, PKC inactivator GF109203X and IP_3_ receptor blocker 2-APB, and Ca^2+^/CaM/CaMK-II blockade using Ca^2+^-free medium, VSCC blocker nefidipine, CaM antagonist calmidazolium, and CaMK-II inhibitor KN62. These findings, taken together, provide evidence that the synergism between IGF-I/-II and TAC3 gene products on SLα gene expression is dependent on TACR3 expression at the pituitary level and subsequent activation of post-receptor signaling cascades functionally coupled with TACR3 activation. 

In mammalian cell models, e.g., pancreatic cancer cells, functional crosstalk between IGF-IR with GPCR signaling has been reported [20] and. in some cases, can be related to enhancement in post-receptor signaling, e.g., for PKC/PKD/MEK signaling linked to G_q_ activation or [Ca^2+^]_i_ mobilization coupled with PI_3_K/Akt/mTOR activation [21,22]. To examine if the synergistic effect on SLα gene expression also involves a functional component with crosstalk/potentiation of post-receptor signaling coupled to IGF-IR and TACR3 activation, IGF-I cotreatment was performed in grass carp pituitary cells together with the respective pharmacological activations for the PKA-, PKC-, and Ca^2+^-dependent pathways known to be coupled with TACR3 activation. In these cases, treatments with the membrane-permeant cAMP analog 8Br.cAMP, AC activator forskolin, DAG analog Dic8, PKC activator TPA, Ca^2+^ ionophore A23187, and VSCC activator BayK8644 were all effective in elevating SLα mRNA expression. However, IGF-I cotreatment could either induce an additive effect or did not trigger further enhancement on SLα gene expression at the pituitary cell level. Given that IGF-IR stimulation is well-documented to induce phosphorylation and subsequent activation of various kinases in the MAPK and PI_3_K/Akt pathways [53,54], the possibility of potentiating IGF-I signaling via TACR3 activation by promoting the phosphorylation status of key kinases for the respective pathways coupled to IGF-IR was also tested. Although the TACR3 agonist senktide did not alter the IGF-I-induced protein phosphorylation of MEK_1/2_, ERK_1/2_, and p_38_MAPK in grass carp pituitary cells, the Akt phosphorylation induced by IGF-I could be notably enhanced by TACR3 activation. These results suggest that the synergistic effect on SLα gene expression by IGFs and TAC3 gene products was also mediated by functional crosstalk of post-receptor signaling between IGF-IR and TACR3 in the grass carp pituitary.

In summary, we have cloned the grass carp TACR3, characterized its receptor binding and signal transduction properties, and confirmed its expression in grass carp pituitary. At the pituitary level, IGF-IR and TACR3 were found to co-localize in SLα cells and IGF-I/-II cotreatment was also effective in triggering a potentiating effect on SLα mRNA expression induced by TAC3 gene products. This potentiating effect could be correlated with the stimulatory actions of IGF-IR activation on TACR3 gene expression in grass carp pituitary cells via activation of the MAPK- and PI_3_K/Akt- pathways. The synergistic effect of IGF-I/-II and TAC3 gene products on SLα gene expression is highly dependent on TACR3 expression at the pituitary level and the post-receptor signaling cascades coupled to TACR3, including the AC/cAMP/PKA, PLC/IP_3_/PKC, and Ca^2+^/CaM/CaMK-II pathways (Figure 7). Furthermore, the potentiating effect of IGF-IR and TACR3 activation could also involve the functional crosstalk at the level of post-receptor signal transduction in grass carp pituitary. Our findings for the first time demonstrate that functional interactions between IGF-I/-II and TAC3 gene products occur in the fish pituitary and play a role in SLα gene expression. 

## Figures and Tables

**Figure 1 cells-08-00887-f001:**
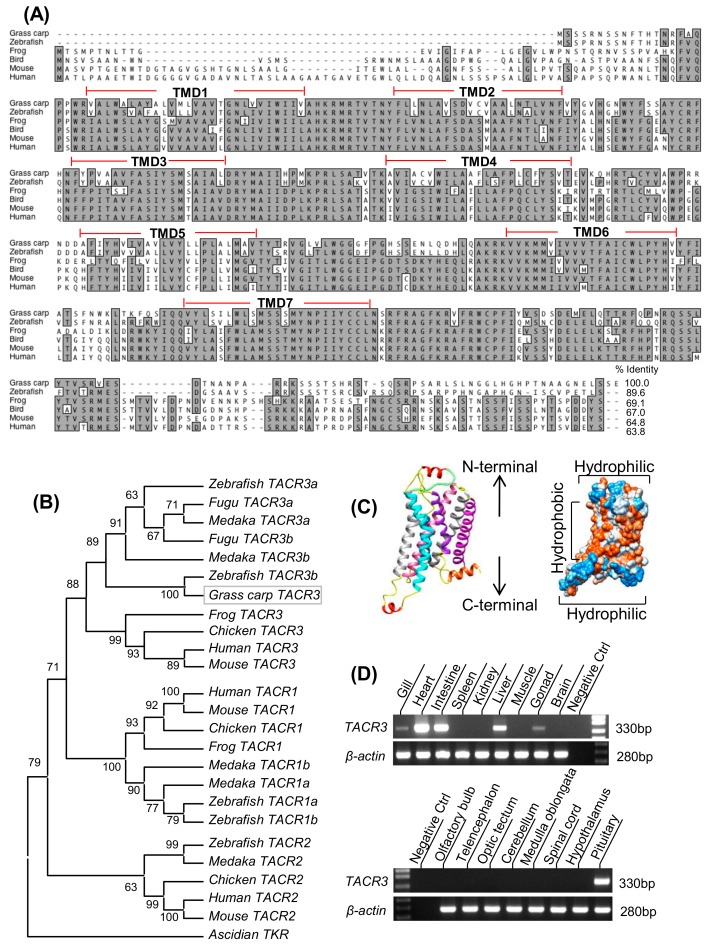
Molecular cloning and tissue distribution of grass carp TACR3: (**A**) Amino acid sequence alignment of grass carp TACR3 using the Clustal-W algorithm with the Mac Vector program. The conserved amino acid (a.a.) residues are boxed in grey, and the seven transmembrane domains are labeled as TMD1–7. (**B**) Phylogenetic analysis of tachykinin receptors nucleotide sequences using neighbor-joining method with MEGA 6.0: Ascidian TKR was used as an out-group. (**C**) Three-dimensional protein model of grass carp TACR3 was deduced based on the crystal structure of human TACR3 using the SWISS-MODEL program. The location of N-terminal and C-terminal were highlighted. (**D**) Tissue distribution profile of grass carp TACR3: Total RNA was isolated from selected tissues and various brain regions in grass carp and subjected to RT-PCR using a primer specific for grass carp TACR3. β-actin was used as the internal control.

**Figure 2 cells-08-00887-f002:**
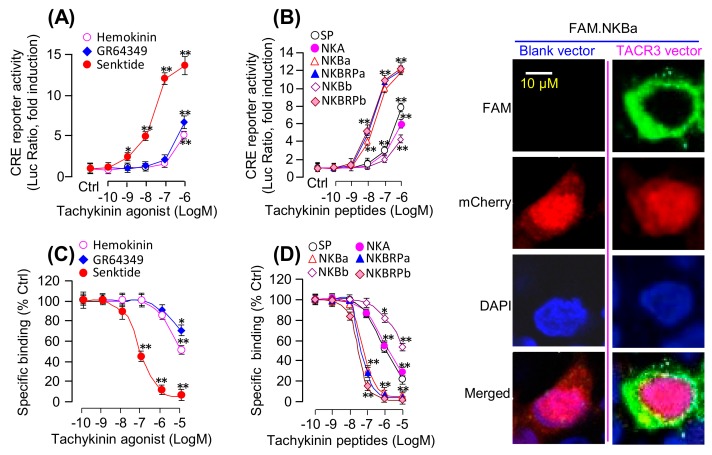
Functional expression and binding affinity analysis of grass carp TACR3: (**A**) HEK-293 cells expressing both grass carp TACR3 and CRE-Luc were treated with various concentrations of TACR1 agonist Hemokinin-1, TACR2 agonist GR64349, and TACR3 agonist senktide (0.1–1000 nM). (**B**) HKE-293 cells with expression of grass carp TACR3 and CRE-Luc were treated with various concentrations of grass carp SP, NKA, NKBa, NKBRPa, NKBb, or NKBRPb (0.1–1000 nM). (**C**) HKE-293 cells stably expressing grass carp TACR3 were pretreated for 30 min with Hemokinin-1, GR64349, or senktide (0.1–1000 nM) prior to adding FAM-NKBRPb (10 nM). (**D**) HEK-293 cells stably expressing grass carp TACR3 were pretreated for 30 min with grass carp SP, NKA, NKBa, NKBRPa, NKBb, or NKBRPb (0.1–1000 nM) prior to adding the FAM-NKBa (10 nM). (**E**) Shown are confocal images of FAM-NKBa bound to HEK-293 cells with blank vector or grass carp TACR3-expressing vector. In this study, HEK-293 cells stably expressing TACR3 were cultured overnight on poly-D-lysine-coated cover glass slides, and then, the cells were incubated with 1 µM FAM-NKBa for 30 min at 37 °C with 5% CO_2_. The asterisk was used to reveal the significant difference between two groups (“*” means *P* < 0.05, and “**” means *P* < 0.01).

**Figure 3 cells-08-00887-f003:**
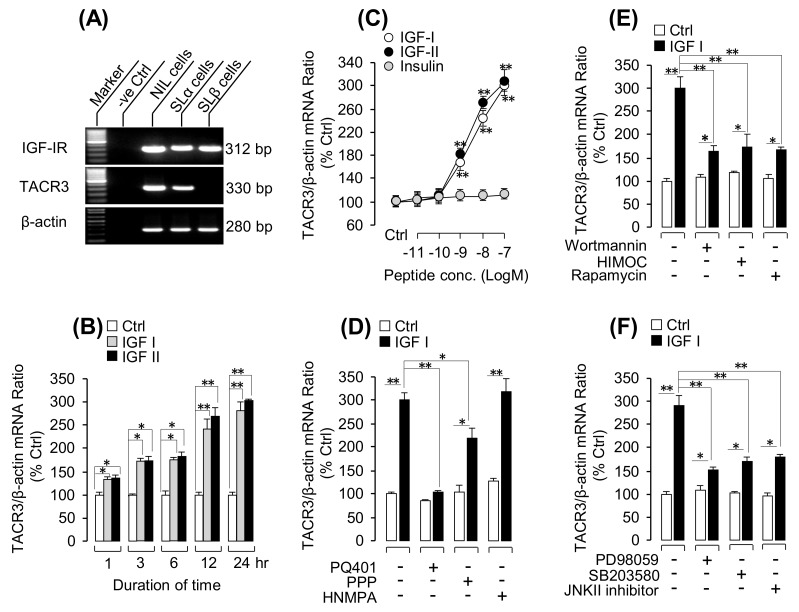
Insulin-like growth factors (IGF)-I/-II-induced TACR3 mRNA expression in grass carp pituitary cells: (**A**) Cell-type specific expression of grass carp TACR3 and IGF-IR in grass carp pituitary cells. By using the laser capture microdissection (LCM) technique, immuno-identified SLα cells and SLβ cells were isolated from grass carp pituitary cells; then, these specific cell groups were subjected into RT-PCR with specific primers for TACR3 and IGF-IR. In this study, RT-PCR of TACR3 and IGF-IR in mixed NIL pituitary cells were used as a positive control while RT-PCR for β-actin was used as internal control. (**B**) Time course of human IGF-I (100 nM) and IGF-II (100 nM) treatment on TACR3 mRNA expression in grass carp pituitary cells; (**C**) dose-dependence of 24-h treatment with increasing levels of IGF-I, IGF-II, and insulin (0.01–100 nM) on TACR3 mRNA expression; and (**D**) receptor specificity for TACR3 regulation by IGF-I in grass carp pituitary cells: In this experiment, grass carp pituitary cells were treated for 24-h with IGF-I (10 nM) in the presence or absence of IGF-IR inhibitor PQ401 (10 µM) or PPP (10 µM) and of insulin receptor inhibitor HNMPA (10 µM), respectively. (**E**) Effects of 24-h cotreatment with the MEK inhibitor PD98059 (10 µM), p_38_MAPK inhibitor SB203580 (10 µM), or JNKII inhibitor (10 µM) on IGF-I (10 nM)-induced TACR3 mRNA expression and (**F**) effects of 24-h cotreatment with the PI_3_K inhibitor wortmannin (1 µM), Akt inhibitor HIMOC (10 µM), and mTOR inhibitor rapamycin (20 nM) on IGF-I (10 nM)-induced TACR3 mRNA expression: The asterisk was used to reveal the significant difference between two groups (“*” means *P* < 0.05, and “**” means *P* < 0.01).

**Figure 4 cells-08-00887-f004:**
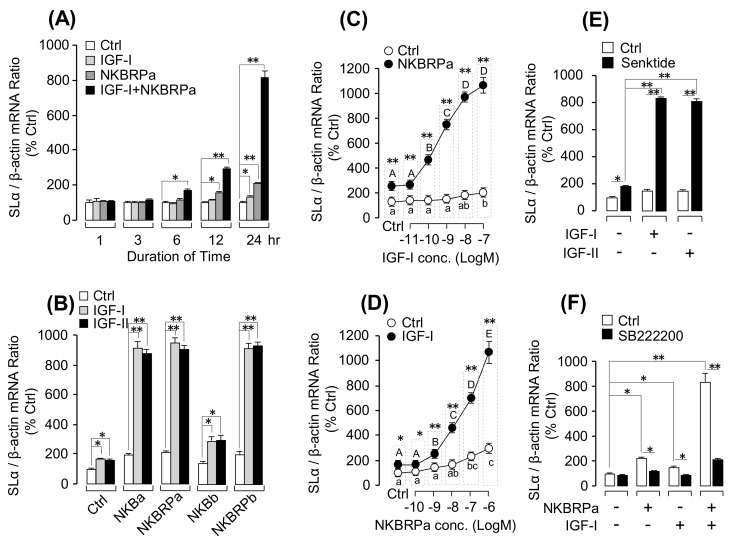
Potentiating effects of IGFs on TAC3 gene product-induced SLα mRNA expression in grass carp pituitary cells: (**A**) Time-course experiment for NKBRPa (1 µM)-, IGF-I (100 nM)-, and NKBRPa (1 µM) + IGF-I (100 nM)–induced SLα mRNA expression; (**B**) synergistic effect of IGF-I/-II with various tachykinins, including NKBa, NKBRPa, NKBb, or NKBRPb (1 µM); (**C**) potentiating effect of IGF-I (0.01–100 nM) on NKBRPa (1 µM)-stimulate grass carp pituitary SLα mRNA expression; (**D**) effect of NKBRPa concentration (0.1–1000 nM) on IGF-I (100 nM)-induced SLα transcript levels in grass carp pituitary cells; (**E**) senktide (1 µM) collaborated with IGF-I/-II (100 nM) to induce grass carp pituitary SLα mRNA expression; and (**F**) grass carp pituitary cells treated with NKBRPa (1 µM), IGF-I (100 nM), and IGF-I (100 nM) + NKBRPa (1 µM) in the presence or absence of TACR3 antagonist SB222200 (10 µM). The asterisk was used to reveal the significant difference between two groups (“*” means *P* < 0.05, and “**” means *P* < 0.01). The different lower-case letters were used to reveal the significant difference between IGF-I or NKB alone and the control group (*P* < 0.05). The different capital letters were used to present the significant difference between IGF-I +NKB and control group (*P* < 0.05).

**Figure 5 cells-08-00887-f005:**
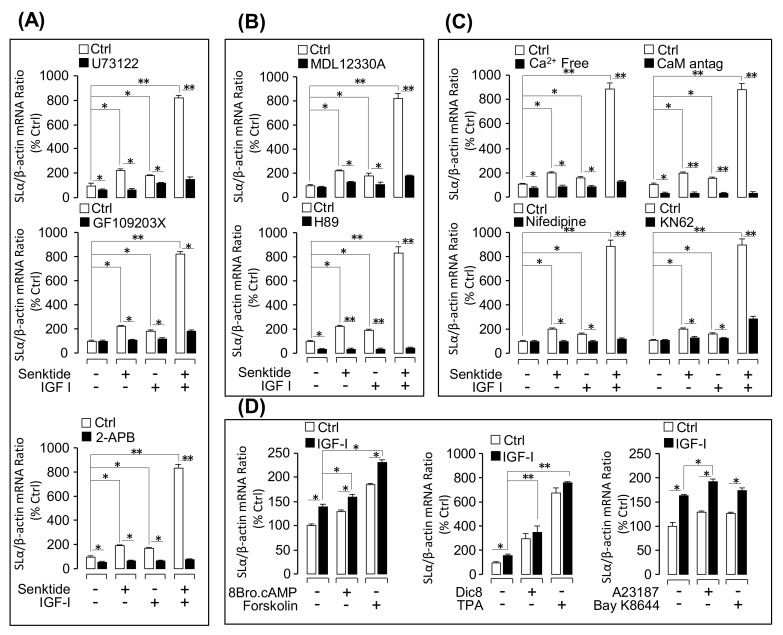
Signal pathways of IGF-I potentiation of senktide-induced SLα mRNA expression: (**A**) Functional role of phospholipase C (PLC)-dependent pathway in the collaborative effect of senktide and IGF-I. In this experiment, U73122 (10 µM), GF109203X (10 µM), or 2-APB (100 µM) were used to block senktide (1 µM), IGF-I (100 nM), and senktide (1 µM)+IGF-I (100 nM)-induced SLα mRNA expression in grass carp pituitary cells. (**B**) MDL12330A (10 µM) or H89 (10 µM) were used to block senktide (1 µM), IGF-I (100 nM), and senktide (1 µM)+IGF-I (100 nM)-induced SLα mRNA expression. (**C**) Ca^2+^-free medium, Nifedipine (10 µM), calmidazolium (1 µM), or CaMK-II inactivator KN62 (10 µM) were used to examine the Ca^2+^ pathway involved in senktide (1 µM), IGF-I (100 nM), and senktide (1 µM)+IGF-I (100 nM)-induced SLα mRNA expression. (**D**) The synergistic effects of IGF-I (100 nM) with 8Br.cAMP (10 µM), forskolin (1 µM), Dic8 (100 µM), TPA (10 µM), A23187 (100 nM), or BayK8644 (100 nM) on SLα mRNA expression are shown. The asterisk was used to reveal the significant difference between two groups (“*” means *P* < 0.05, and “**” means *P* < 0.01).

**Figure 6 cells-08-00887-f006:**
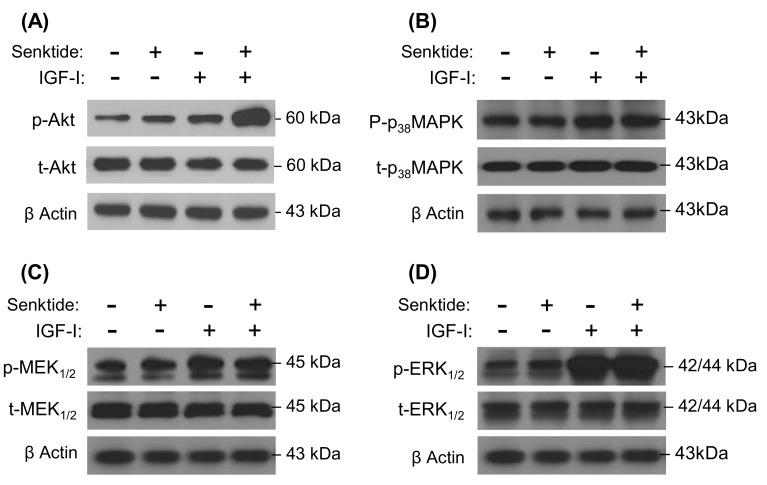
Effects of senktide, IGF-I, and senktide + IGF-I on protein phosphorylation and total protein production of Akt, p_38_MAPK, MEK_1/2_, and ERK_1/2_ in grass carp pituitary cells: Senktide (1 µM), IGF-I (50 nM), or senktide (1 µM) + IGF-I (50 nM) were used to incubate grass carp pituitary cells for 15 min. After that, cell lysate was prepared, size fractionated by SDS-PAGE, and subjected to Western blot using the antibodies for phosphorylated form (p-) and total protein (t-) of MEK_1/2_, ERK_1/2_, Akt, and p_38_MAPK. In these experiments, parallel blotting for β-actin expression was routinely performed to serve as a loading control for Western blot.

**Figure 7 cells-08-00887-f007:**
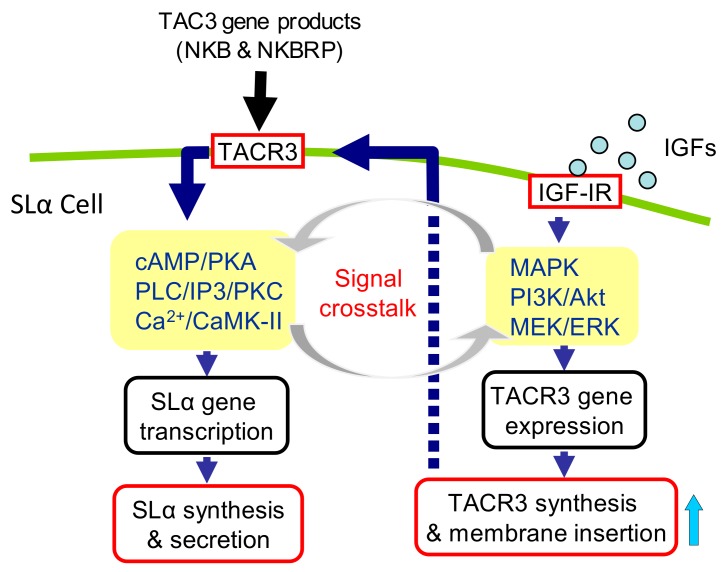
Working model of the potentiating effect of IGF-I/-II on TAC3 gene product-induced SLα mRNA expression in grass carp pituitary: In grass carp, both TACR3 and IGF-IR were expressed in pituitary SLα cells. In grass carp SLα cells, IGF-I/-II could stimulate TACR3 mRNA expression via IGF-IR activation by functional coupling with the PI_3_K/Akt/mTOR, MKK/MAPK, and MEK_1/2_/ERK_1/2_ pathways. The stimulation of TACR3 gene expression could contribute to the potentiating effect of IGFs on NKBRPa- or NKBa-induced SLα mRNA expression, which were mediated by TACR3 via the PLC/IP_3_/PKC, AC/cAMP/PKA, and Ca^2+^/CaM/CaMK-II signaling pathways. Furthermore, the potentiating effect of IGF-IR and TACR3 activation could also involve the functional crosstalk at the level of post-receptor signal transduction in grass carp pituitary.

**Table 1 cells-08-00887-t001:** ED_50_ values of the bioactivities of grass carp TAC3 and TAC1 gene products via NK3R activation.

Tachykinins	TAC1		TAC3a		TAC3b	
Luciferase Reporters	SP	NKA	NKBa	NKBRPa	NKBb	NKBRPb
**ED_50_ (nM) for NK3R-mediated luciferase activity**						
CRE-Luc	701.8 ± 50.1	1512.3 ± 25.1	26.9 ± 3.2	17.9 ± 2.5	>10 µM	15.9 ± 2.9
AP1-Luc	954.1 ± 33.3	719.2 ± 39.3	36.5 ± 7.8	27.5 ± 5.3	>10 µM	27.1 ± 2.3
NFAT-Luc	1071.3 ±65.4	1154.4 ± 30.8	37.1 ± 8.1	30.1 ± 2.2	>10 µM	23.9 ± 6.2
SRE-Luc	1060.2 ± 34.8	752.8 ± 20.5	15.5 ± 1.5	22.3 ± 5.8	>10 µM	12.7 ± 3.8
**ED_50_ (nM) for Ca^2+^ responses under Ca^2+^-containing/Ca^2+^-free condition**						
Ca^2+^-free	492.4 ± 48.2	480.8 ± 26.5	31.8 ± 3.6	38.6 ± 2.8	>10 µM	40.4 ± 7.8
Ca^2+^-containing	487.1 ± 37.7	392.3 ± 23.4	50.3 ± 8.4	31.7 ± 4.9	>10 µM	33.2 ± 3.8

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
