# Peer review of "IGFs Potentiate TAC3-induced SLα Expression via Upregulation of TACR3 Expression in Grass Carp Pituitary Cells"

_cells, 2019, doi:10.3390/cells8080887_

Round 1
Reviewer 1 Report
Letter to Authors
cells-562424
IGFs Potentiation of TAC3-induced SLα Expression via Up-regulation of TACR3 Expression in the Pituitary Cells
Guangfu Hu, Anderson O. L. Wong, Mulan He, Wendy K. W. KO
Dear authors,
It was a pleasure of me to read this interesting MS. Your MS describes a pivotal role of TACR3 in a signal transduction cascade to SL synthesis pathway in a teleost fish in addition to your previous works that described the IGF- and TAC3-SL pathways. This MS is good for publication in the journal you are submitting to. Before publication, you need a round of minor revision mostly to help readers' understanding. See below for detail. I am looking forward to seeing your interesting work in the journal.
General Comments
I think it is better to put the abbreviations at the top of the main text following keywords or as footnotes at the first page to help readers' understanding. Consult with a layout editor of the journal.
Citation in the main text should be in brackets.
Give suppliers' information of chemicals and instruments consistently. You seem to give supplier's name and address at the first place and cut address off later, and this is nice, but not consistent at the current delivery.
L17,19,83,91,etc
carp -> grass carp
A bare word 'carp' mens Cyprinus carpio, a tetraploid teleost. At the introduction section, readers will misunderstand that your cloned gene product was tested on this tetraploid fish pituitary cell with excess paralogs, and wonder how the author sorted out these paralogs.
L28
TACR3, IGFs, SLalpha, pituitary -> delete
Replace with other words that do not appear in the title to draw wider readership avoiding good-for-nothing duplicate hits upon computer search. Hint; signal transduction, cascade, pharmacological test, agonist, antagonist, tachykinin receptor, etc.
L35
the field?
What field is it? Reproductive biology?
L74
SP?
Spell-out is absent from the abbreviation section.
L82
HEK-293 -> human HEK-293
L103
MD -> MO
L106
St. Louis, Missouri -> delete
L109,111
Table 1 should be a supplement table. Renumbering the supplement is needed accordingly.
L110
Maryland -> MD
L116
Invitrogen -> Invitrogen (Carlsbad, CA)
L118
Full-length cDNA -> Full-length cDNA obtained using 5'/3' RACE
L129
PT3286-1 -> delete
L133
pcDNA3.1/Zeo(-) -> pcDNA3.1/Zeo(-) (Invitrogen)
L136
Add supplier information of OPTI-MEM.
137
Promega -> Promega (Madison, WI)
L143,144
BJ -> delete
L150
San Diego, CA -> delete
Address information is unnecessary at the second place and later.
L152
Cat. No.: A-6823 -> Eugene, OR
Address information is needed rather than Cat. No. consistently with others.
L155
MA -> Waltham, MA
L159
MA -> delete
L162
Add address information of BMG Labtechonolies.
L174
Fluoresce -> Carboxyfluorescein?
L191
Carl Zeiss -> Carl Zeiss, Oberkochen, Germany
L195
SLalpha cells?
SLalpha-{producing, secreting, expressing} cells?
L203
with primers specific for the respective receptor subtypes (Supplemental Table 2) (redundant) -> as above
L210
total RNA was extracted from individual well using Trizol and reversely transcribed by Superscript II (50 Unit, 211 Invitrogen) (redundant) -> total RNA was extracted from individual well and reversely transcribed as above
L213
specific for grass carp SLalpha and TACR3 (Supplemental Table 2), respectively (redundant) -> mentioned above
L225
cells lysate -> cell lysate
L228,231
Supplier information is necessary on primary and secondary antibodies. When you used your home-made antisera, cite your work properly.
L230
Washing buffer should be addressed. PBS, TBS-T or other?
L233
Rockford -> Rockford, IL
L234
Eastman Kodak -> Eastman Kodak, Rochester, NY
L254
Using 5'/3' RACE -> delete
See L118.
L258
using SWISS-MODEL program (redundant) -> delete
See L121.
L277
select -> selected
L463
Beta actin blot in Fig.6C&D seem identical except for presence (C)/absence (D) of tiny blobs near ++ lane. Is it OK?
L481
highly -> delete
See L255.
L569-571
To examine if .. IGF-IR and TACR3 activation.
Not a complete sentence. A comma?
L610
Do abbreviations stand for genes and their products only? How about adding other abbreviations such as for FAM, NIL instead of L195, etc? Also how about other abbreviations like DAG, GPCR, Gq, mTOR, PACAP, SP, VSCC, etc? Very familier cAMP, cDNA, PCR, etc are OK without spell-out.
L614
SLalpha, Somatolactin alpha -> SL, Somatolactin
Or otherwise, you may present both SLalpha and beta, but this is verbose.
L615
VSCC?
L626 references
Check the list again carefully from the beginning. You might added, omitted swapped references upon internal revision processes. Shift in numbering might occurred, and if so, readers will think your citation is irrelevant. It is the authors' responsibility that all references are properly cited.
Author Response
Dear authors,
It was a pleasure of me to read this interesting MS. Your MS describes a pivotal role of TACR3 in a signal transduction cascade to SL synthesis pathway in a teleost fish in addition to your previous works that described the IGF- and TAC3-SL pathways. This MS is good for publication in the journal you are submitting to. Before publication, you need a round of minor revision mostly to help readers' understanding. See below for detail. I am looking forward to seeing your interesting work in the journal.
Response:
We thank the reviewer for her/his praise and suggestions on our manuscript. We also realized the shortcoming on our manuscript. As you suggested, we have tried our best to revise the manuscript again.
General Comments
I think it is better to put the abbreviations at the top of the main text following keywords or as footnotes at the first page to help readers' understanding. Consult with a layout editor of the journal.
Response:
Thank you very much for your suggestions. Following your suggestions, we have list the full name of the abbreviations on their first occurrence. Since this journal prefer putting the abbreviations at the back of the main text, our abbreviations were also listed in the back of the main text. (Line 614-622)
Abbreviations: AC: Adenylyl cyclase; Akt, protein kinase B; CaM, Calmodulin; CaMK-II, CaM-dependent protein kinase II; DAG, diacylglycerol; ERK, extracellular signal-regulated kinase; FAM, fluorescein amidite; GPCR, G protein coupled receptor; Gq, G-protein alpha subunit, group Q; IGF, insulin-like growth factor; IP3, Inositol 1,4,5-triphosphate; LCM, laser capture microdissection; MAPK, mitogen-activated protein kinase; mTOR, mammalian target of rapamycin; TACR3, tachykinin receptor 3; NIL, neurointermediate lobe, NKB, neurokinin B; NKBRP, NKB-related peptide; PACAP, pituitary adenylate cyclase activating peptide; PI3K, phosphatidylinositol-3-kinase; PKC, Protein kinase C; PKA, Protein kinase A; PLC, phospholipase C; RLU, relative luminescence unit; SL, Somatolactin; SP, Substance P; TAC3, tachykinin 3; TACR3, tachykinin receptor 3; VSCC, voltage sensitive calcium channel.
Citation in the main text should be in brackets.
Give suppliers' information of chemicals and instruments consistently. You seem to give supplier's name and address at the first place and cut address off later, and this is nice, but not consistent at the current delivery.
Response:
Thank you very much for your comments. As you suggested, the suppliers’ information of chemical and instruments have been added. (Line 122-183)
L17,19,83,91,etc
carp -> grass carp
A bare word 'carp' mens Cyprinus carpio, a tetraploid teleost. At the introduction section, readers will misunderstand that your cloned gene product was tested on this tetraploid fish pituitary cell with excess paralogs, and wonder how the author sorted out these paralogs.
Response:
Thank you very much for your comments. As you suggested, we have replaced the “carp” with grass carp in the full manuscript. (Line 17, 19, 98, 100, 139…..)
L28
TACR3, IGFs, SLalpha, pituitary -> delete
Replace with other words that do not appear in the title to draw wider readership avoiding good-for-nothing duplicate hits upon computer search. Hint; signal transduction, cascade, pharmacological test, agonist, antagonist, tachykinin receptor, etc.
Response:
Thank you very much for your suggestion. The keywords have been replaced with “signal transduction; tachykinin receptor; pharmacological test; neurokinin B; grass carp”. (Line 33)
L35
the field?
What field is it? Reproductive biology?
Response:
Thank you very much for your comment. As you suggested, the field was replaced with “reproductive biology field”. (Line 41)
L74
SP?
Spell-out is absent from the abbreviation section.
Response:
Sorry for the shortcoming on our side. The “SP” has been replaced with “Substance P (SP)”. (Line 79, 621)
L82
HEK-293 -> human HEK-293
Response:
Thank you very much for your comment. As you suggested, the HKE-293 was replaced with human HEK-293. (Line 99)
L103
MD -> MO
Response:
Sorry for our shortcoming in this size. Typo errors have been corrected accordingly.(Line 122)
L106
St. Louis, Missouri -> delete
Response:
Thank you very much for your comment. As you suggested, the St. Louis, Missouri has been deleted in the revised manuscript. (Line 125)
L109,111
Table 1 should be a supplement table. Renumbering the supplement is needed accordingly.
Response:
We thank the reviewer for her/his suggestion. As you suggested, the Table 1 has been listed as the Supplemental Table 2 in the revised manuscript. We have also renumbered the supplemental tables. (Line 128, 140)
L110
Maryland -> MD
Response:
Thank you very much for your comment. As you suggest, the Maryland has been replaced with MD. (Line 129)
L116
Invitrogen -> Invitrogen (Carlsbad, CA)
Response:
Thank you very much for your comment. As you suggest, the Invitrogen has been replaced with Invitrogen (Carlsbad, CA). (Line 132)
L118
Full-length cDNA -> Full-length cDNA obtained using 5'/3' RACE
Response:
Thank you very much for your suggestion. The “full-length cDNA” has been revised as the “Full-length cDNA obtained using 5'/3' RACE”. (Line 134)
L129
PT3286-1 -> delete
Response:
Thank you very much for pointing this. The PT3286-1 has been deleted in the revised manuscript. (Line 146)
L133
pcDNA3.1/Zeo(-) -> pcDNA3.1/Zeo(-) (Invitrogen)
Response:
Thank you very much for your suggestion. The “pcDNA3.1/Zeo(-)” has been revised as the “pcDNA3.1/Zeo(-) (Invitrogen, Carlsbad, CA)”. (Line 150)
L136
Add supplier information of OPTI-MEM.
Response:
Thank you very much for pointing this. The supplier information of OPTI-MEM has been added in the Line 154.
137
Promega -> Promega (Madison, WI)
Response:
Thank you very much for pointing this. The address information for Promega was added in Line 155.
L143,144
BJ -> delete
Response:
Thank you very much for your comment. As you suggest, the BJ has been deleted in Line 162.
L150
San Diego, CA -> delete
Address information is unnecessary at the second place and later.
Response:
Thank you very much for your comment. As you suggest, the San Diego, CA has been deleted in Line 167.
L152
Cat. No.: A-6823 -> Eugene, OR
Address information is needed rather than Cat. No. consistently with others.
Response:
Thank you very much for your suggestion. The Cat. No. was replaced with address information (Eugene, OR) in Line 169.
L155
MA -> Waltham, MA
Response:
Thank you very much for your comment. As you suggest, the address information was added in Line 172.
L159
MA -> delete
Response:
Thank you very much for your comment. As you suggest, the MA has been deleted in Line 176.
L162
Add address information of BMG Labtechonolies.
Response:
Thank you very much for your suggestion. The address information of BMG was added in Line 179 (BMG Labtech, Ortenberg).
L174
Fluoresce -> Carboxyfluorescein?
Response:
Thank you very much for your comment. You are right. The FAM Carboxyfluorescein was more suitable. (Line 191)
L191
Carl Zeiss -> Carl Zeiss, Oberkochen, Germany
Response:
Thank you very much for your suggestion. The address information of Carl Zeiss was added in Line 208.
L195
SLalpha cells?
SLalpha-{producing, secreting, expressing} cells?
Response:
Thank you very much for your comment. It should be the SLα producing cells (Line 213).
L203
with primers specific for the respective receptor subtypes (Supplemental Table 2) (redundant) -> as above
Response:
Thank you very much for your suggestion. The redundant part has been replaced with “as above”. (Line 222)
L210
total RNA was extracted from individual well using Trizol and reversely transcribed by Superscript II (50 Unit, 211 Invitrogen) (redundant) -> total RNA was extracted from individual well and reversely transcribed as above
Response:
Thank you very much for your suggestion. The redundant part has been deleted in Line 228-229.
L213
specific for grass carp SLalpha and TACR3 (Supplemental Table 2), respectively (redundant) -> mentioned above
Response:
Thank you very much for your suggestion. The redundant part has been replaced with “mentioned above”. (Line 231)
L225
cells lysate -> cell lysate
Response:
Thank you very much for your comment. The modification has been made in Line 243.
L228,231
Supplier information is necessary on primary and secondary antibodies. When you used your home-made antisera, cite your work properly.
Response:
Thank you very much for your comment. The supplier information of primary and secondary antibodies have been listed in the “Animal and regents” and Supplemental Table 2 (Line 127-129).
L230
Washing buffer should be addressed. PBS, TBS-T or other?
Response:
Thank you very much for your comment. The washing buffer is TBS-T. (Line 248)
L233
Rockford -> Rockford, IL
Response:
Thank you very much for your comment. The modification has been made in Line 251.
L234
Eastman Kodak -> Eastman Kodak, Rochester, NY
Response:
Thank you very much for your comment. The address information has been added in Line 252.
L254
Using 5'/3' RACE -> delete
See L118.
Response:
Thank you very much for your suggestion. The redundant information (Using 5'/3' RACE) has been deleted in Line 274.
L258
using SWISS-MODEL program (redundant) -> delete
See L121.
Response:
Thank you very much for your suggestion. The redundant information has been deleted in Line 278.
L277
select -> selected
Response:
Sorry for our shortcoming in this size. Typo errors have been corrected accordingly.(Line 294)
L463
Beta actin blot in Fig.6C&D seem identical except for presence (C)/absence (D) of tiny blobs near ++ lane. Is it OK?
Response:
Thank you very much for your comment. The question for the same beta actin in Fig.6C&D, it seems that the Western blot for MEK & ERK were from the same experiment. That is the reason why they shared the same internal control.
L481
highly -> delete
See L255.
Response:
Thank you very much for your suggestion. The redundant information (highly) has been deleted in Line 495.
L569-571
To examine if .. IGF-IR and TACR3 activation.
Not a complete sentence. A comma?
Response:
Thank you very much for your comment. The full stop has been replaced with a comma in Line 583-585.
L610
Do abbreviations stand for genes and their products only? How about adding other abbreviations such as for FAM, NIL instead of L195, etc? Also how about other abbreviations like DAG, GPCR, Gq, mTOR, PACAP, SP, VSCC, etc? Very familier cAMP, cDNA, PCR, etc are OK without spell-out.
Response:
Thank you very much for your suggestion. More abbreviations have been added in Line 614-624.
NIL, neurointermediate lobe; FAM, fluorescein amidite; LCM, laser capture microdissection; DAG, diacylglycerol; GPCR, G protein coupled receptor; mTOR, mammalian target of rapamycin; SP, Substance P; PACAP, pituitary adenylate cyclase activating peptide; VSCC, voltage sensitive calcium channel; Gq, G-protein alpha subunit, group Q;
L614
SLalpha, Somatolactin alpha -> SL, Somatolactin
Or otherwise, you may present both SLalpha and beta, but this is verbose.
Response:
Thank you very much for your suggestion. The modification has been made in Line 631.
L615
VSCC?
Response:
Sorry for our shortcoming in this size. The abbreviation of VSCC has been listed in Line 632.
L626 references
Check the list again carefully from the beginning. You might added, omitted swapped references upon internal revision processes. Shift in numbering might occurred, and if so, readers will think your citation is irrelevant. It is the authors' responsibility that all references are properly cited.
Response:
Thank you very much for your suggestion. We have check the references again carefully.
One reference has been added in the revised version.
Hu, G.F.; He, M.L.; Wong, A.O.L. Novel functional role of NK3R expression in the potentiating effects on somatolactin α autoregulation in grass carp pituitary cells. Scientific Report, 2016, 6: 36102.
Reviewer 2 Report
In the manuscript entitled “IGFs Potentiation of TAC3-induced SLα Expression via Up-regulation of TACR3 Expression in the Pituitary Cells”, Hu et al. demonstrate that in grass carp pituitary cells, IGF-I and IGF-II synergize with TAC3 gene products to induce SLα expression through upregulation of TACR3”.
The major findings/achievements of this manuscript are:
1) Grass carp TACR3 was cloned and ligand binding and signaling pathways activated were characterized;
2) TACR3 is expressed in grass carp pituitary cells, namely somatolactin α (SLα)-positive cells;
3) IGF-IR is also expressed in SLα-positive grass carp pituitary cells;
4) IGF-IR induces TACR3 gene expression in carp pituitary cells via activation of MAPK and PI3K/Akt pathways;
5) IGF-I/-II and TAC3 gene products synergize to activate SLα gene expression in grass carp pituitary cells;
6) This synergism was shown to be dependent on TACR3 expression and post-receptor signaling coupled to TACR3, including the AC/cAMP/PKA, PLC/IP3/PKC and Ca2+/CaM/CaMK-II.
This study follows a previous report (Hu et al. 2014, Endocrinology) in which the same group characterized TAC3 gene products in terms of receptor specificity and signal transduction for prolactin and SLα regulation in pituitary cells, using the same fish model, and a second report (Hu et al. 2016, Scientific Reports) showing that local release of SLα and SLβ could interact with IGF-I/-II and TAC3 gene products to up-regulate SLα gene expression via stimulation of TACR3 expression at the pituitary level. Here, the authors deepen the study of IGFs, TAC3 products and SLα interactions.
The main analyses are appropriate and the conclusions of the authors are supported by the results. However, I have some suggestions to improve the manuscript.
1) The Introduction could be improved. Firstly, the article Hu et al. 2016, “Novel Functional Role of NK3R Expression in the Potentiating Effects on Somatolactin α Autoregulation in grass carp pituitary cells”, published on Scientific Reports, should be cited in order to show the authors’ previous results concerning the same theme. In addition, the motives that led the authors to follow their research the way they did should be mentioned and, in this context, the novelty of this new study should be stated. Secondly, the authors are not clear in showing the interest and/or the intended practical application of the research and the reason for using this particular animal model. So, these matters should be clarified.
2) I have two critiques regarding the Methods. First, although it is mentioned that “no significant changes were noted for β-actin mRNA levels between different experiment groups”, qPCR results should be normalized anyway. Second, regarding statistics, the authors always apply ANOVA but they do not refer if they tested for normality and equality of variances before to justify the use of ANOVA instead of the alternative nonparametric test.
3) Although the English language is understandable, it should be improved by doing a spell/grammar check. For instance, in line 109, where “second antibodies” is written, it should be written “secondary antibodies” instead, and in line 226, where “and the clear supernatant were resolved” is written, it should be “and the clear supernatant was resolved” instead. There are also some excerpts where the meaning is unclear and ambiguous or difficult to follow due to long phrases and the lack of punctuation (or misused punctuation). One example is the phrase contained in lines 36 to 40.
4) The Supplementary Tables and Figures were not revised because I did not have access to them.
Specific comments:
1. In my opinion the title is somewhat confusing and should provide more information, namely concerning the animal model used. I suggest a title like “IGFs potentiate TAC3-induced SLα expression via upregulation of TACR3 in grass carp pituitary cells”.
2. I do not follow why the term “simply” is used two times (lines 242 and 244) to describe the analyses and presentation of data in the Methods section. I think the term should be removed because it gives the idea that something else should have been done but the authors decided to do something that was simpler. Was it the case?
3. In terms of analytic and biological replicates, the authors do not give essential detail. For instance, the analyses shown in Figures 1D, 3A and 6 were done for how many carps: only one or more than one? In the latter case, what is shown (Fig.1D, 3A and 6) is representative of the various analyses? Also, when grass carp pituitary primary cell cultures were used in experiments with several treatments, was each experiment repeated using cells collected from at least one more carp? At least two independent experiments showing essentially the same results should have been done. And, in each experiment, how many replicates were done for each treatment (i.e., the mean ± SEM refers to how many analyses)? I do not understand what is meant by “the data presented (expressed as Mean ± SEM) were pooled results from 6-8 experiments” (lines 249 and 250). Could it be clarified?
4. Table 1 (letters too small, too much information) and Figure 1D (the name of the organs/tissues analyzed) have poor quality and Figure 2B and D are difficult to interpret (the lines are overlapped). Regarding also Figure 2B and D, the symbols and colors used for each peptide should be maintained in both graph because it gets easier to interpret and compare the results.
5. For Figures 1D and 3A it should be indicated in the legend what is “-ve”. I guess is the negative control but it should be indicated.
6. The authors use letters like a, b, c, d, e, f in Figures 2, 3, 4 and 5 to label statistical significance (instead of stars). They should provide the meaning in the figure legends; I noticed that in Figure 3, 4 and 5, the authors refer “the groups denoted by different letters represent a significant difference at p<0.05 (ANOVA followed by Dunnett’s test)” but I think it is not clear enough. It took me some time to understand, especially in Fig 2, where nothing is referred in the legend.
Author Response
Comments and Suggestions for Authors
In the manuscript entitled “IGFs Potentiation of TAC3-induced SLα Expression via Up-regulation of TACR3 Expression in the Pituitary Cells”, Hu et al. demonstrate that in grass carp pituitary cells, IGF-I and IGF-II synergize with TAC3 gene products to induce SLα expression through upregulation of TACR3”.
The major findings/achievements of this manuscript are:
1) Grass carp TACR3 was cloned and ligand binding and signaling pathways activated were characterized;
2) TACR3 is expressed in grass carp pituitary cells, namely somatolactin α (SLα)-positive cells;
3) IGF-IR is also expressed in SLα-positive grass carp pituitary cells;
4) IGF-IR induces TACR3 gene expression in carp pituitary cells via activation of MAPK and PI3K/Akt pathways;
5) IGF-I/-II and TAC3 gene products synergize to activate SLα gene expression in grass carp pituitary cells;
6) This synergism was shown to be dependent on TACR3 expression and post-receptor signaling coupled to TACR3, including the AC/cAMP/PKA, PLC/IP3/PKC and Ca2+/CaM/CaMK-II.
This study follows a previous report (Hu et al. 2014, Endocrinology) in which the same group characterized TAC3 gene products in terms of receptor specificity and signal transduction for prolactin and SLα regulation in pituitary cells, using the same fish model, and a second report (Hu et al. 2016, Scientific Reports) showing that local release of SLα and SLβ could interact with IGF-I/-II and TAC3 gene products to up-regulate SLα gene expression via stimulation of TACR3 expression at the pituitary level. Here, the authors deepen the study of IGFs, TAC3 products and SLα interactions.
The main analyses are appropriate and the conclusions of the authors are supported by the results. However, I have some suggestions to improve the manuscript.
Response:
We thank the reviewer for her/his praise and suggestions on our manuscript. We also realized the shortcoming on our manuscript. As you suggested, we have tried our best to revise the manuscript again.
1) The Introduction could be improved. Firstly, the article Hu et al. 2016, “Novel Functional Role of NK3R Expression in the Potentiating Effects on Somatolactin α Autoregulation in grass carp pituitary cells”, published on Scientific Reports, should be cited in order to show the authors’ previous results concerning the same theme. In addition, the motives that led the authors to follow their research the way they did should be mentioned and, in this context, the novelty of this new study should be stated. Secondly, the authors are not clear in showing the interest and/or the intended practical application of the research and the reason for using this particular animal model. So, these matters should be clarified.
Response:
Thank you very much for your suggestions. As you suggested, the Introduction has been revised as follows.
(1) Firstly, the article Hu et al. 2016, “Novel Functional Role of NK3R Expression in the Potentiating Effects on Somatolactin α Autoregulation in grass carp pituitary cells”, published on Scientific Reports, should be cited in order to show the authors’ previous results concerning the same theme. In addition, the motives that led the authors to follow their research the way they did should be mentioned and, in this context, the novelty of this new study should be stated.
Response for Q(1):
“Recently, our lab found that co-treatment with somatolactins, IGFs and TAC3 gene product together could serve as a highly potent stimulatory signal for SLα mRNA expression. Further study found that somatolactin could enhance the TAC3 gene products-induced SLα mRNA expression via up-regulation of TACR3 expression (26). However, the reason why IGF could further enhance the potentiating effects of somatolactin on TAC3 gene products-induced pituitary SLα mRNA expression was not clear.”(Line 82-87)
(2) Secondly, the authors are not clear in showing the interest and/or the intended practical application of the research and the reason for using this particular animal model. So, these matters should be clarified.
Response for Q(2):
Thank you very much for your suggestion. As you suggested, we try to clear the intended practical application of the research and the reason for using this particular animal model.
“In China, grass carp (Ctenopharyngodon idellus) is the most important aquaculture species, whose total production has reached 5.50 million tonnes in 2018. Generally, grass carp need to take about five years to reach sexual maturity, which severely restricts the selection of breeding processes. Considered high market demands and long-term growth periods of grass carp, revealing the regulatory mechanisms in sexual maturation of grass carp will be crucial for grass carp breeding. In mammals, previous studies have demonstrated that TAC3/TACR3 system played pivotal roles in the regulation of reproduction and puberty onset (3). Our recent studies have also found that TAC3/TACR3 system should also play an important role in grass carp puberty onset via regulation of pituitary SLα secretion and synthesis (8). Now, we would like to know whether other factors could enhance TAC3/TACR3 system-regulated puberty onset in grass carp.” (Line 88-97)
2) I have two critiques regarding the Methods. First, although it is mentioned that “no significant changes were noted for β-actin mRNA levels between different experiment groups”, qPCR results should be normalized anyway. Second, regarding statistics, the authors always apply ANOVA but they do not refer if they tested for normality and equality of variances before to justify the use of ANOVA instead of the alternative nonparametric test.
Response:
Thank you very much for your comments. As you suggested, the qPCR results have been normalized with β-actin mRNA levels (as shown in revised Figures). In addition, as you suggested, the statistic analysis have also been revised again.
“The six replicates presented (expressed as Mean ± SEM) were pooled results from two individual experiments prior to statistical analysis, all data were tested for normality of distribution using the Shapiro-Wilk normality test. Then one-way ANOVA or two-way ANOVA were used to test the significant difference according to different experiment groups. The differences between groups were considered as significant at P<0.05 (“*”) or highly significant at P<0.01 (“**”).” (Line 265-270)
3) Although the English language is understandable, it should be improved by doing a spell/grammar check. For instance, in line 109, where “second antibodies” is written, it should be written “secondary antibodies” instead, and in line 226, where “and the clear supernatant were resolved” is written, it should be “and the clear supernatant was resolved” instead. There are also some excerpts where the meaning is unclear and ambiguous or difficult to follow due to long phrases and the lack of punctuation (or misused punctuation). One example is the phrase contained in lines 36 to 40.
Response:
Thank you very much for your comments. Sorry for our shortcoming. The “second antibodies” has been replaced with “secondary antibodies”. (Line 128)
The “and the clear supernatant were resolved” was replaced with “and the clear supernatant was resolved”. (Line 243-244)
The long phrases have been divided into two short sentences as follows.
“The study of TAC3/TACR3 system has been recently extended to fish models, e.g., zebrafish (5-7) and grass carp (8). Interestingly, unlike the situation in mammals, the TAC3 gene in fish not only encodes the mature peptide of NKB but also the a.a. sequence of a novel tachykinin-like peptide named NKB-related peptide (NKBRP).”(Line 43-46)
4) The Supplementary Tables and Figures were not revised because I did not have access to them.
Response:
Sorry for our shortcoming. The three Supplemental Tables and two supplementary figures have been added in the revised manuscript.
Specific comments:
In my opinion the title is somewhat confusing and should provide more information, namely concerning the animal model used. I suggest a title like “IGFs potentiate TAC3-induced SLα expression via upregulation of TACR3 in grass carp pituitary cells”.
Response:
Thank you very much for your suggestion. As you suggested, the title has been replaced with “IGFs potentiate TAC3-induced SLα expression via upregulation of TACR3 in grass carp pituitary cells”. (Line 2-4)
I do not follow why the term “simply” is used two times (lines 242 and 244) to describe the analyses and presentation of data in the Methods section. I think the term should be removed because it gives the idea that something else should have been done but the authors decided to do something that was simpler. Was it the case?
Response:
Thank you very much for your comment. The term “simply” has been deleted in Line 258 and Line 261.
In terms of analytic and biological replicates, the authors do not give essential detail. For instance, the analyses shown in Figures 1D, 3A and 6 were done for how many carps: only one or more than one? In the latter case, what is shown (Fig.1D, 3A and 6) is representative of the various analyses? Also, when grass carp pituitary primary cell cultures were used in experiments with several treatments, was each experiment repeated using cells collected from at least one more carp? At least two independent experiments showing essentially the same results should have been done. And, in each experiment, how many replicates were done for each treatment (i.e., the mean ± SEM refers to how many analyses)? I do not understand what is meant by “the data presented (expressed as Mean ± SEM) were pooled results from 6-8 experiments” (lines 249 and 250). Could it be clarified?
(1) For instance, the analyses shown in Figures 1D, 3A and 6 were done for how many carps: only one or more than one? In the latter case, what is shown (Fig.1D, 3A and 6) is representative of the various analyses?
Response:
Thank you very much for your comments.
In Figure 1D, three grass carps were recruited to do the tissue distribution. Each organ/tissue was collected from each grass carp. Total RNA was isolated from each tissue and quantified by Nanodrop 2000. Then the equal RNA from the same three tissues were mixed together to be reversely transcribed.
In Figure 3A, to confirm the co-expression of TACR3 and IGF-IR in SLα producing cells, the neurointermediate lobe (NIL) with high level of SL cells was manually dissected from twelve grass carp pituitaries. (Line 214)
In Figure 6, the pituitary cells derived from ten grass carp pituitaries were used for WB analysis. (Line 236)
(2) Also, when grass carp pituitary primary cell cultures were used in experiments with several treatments, was each experiment repeated using cells collected from at least one more carp?
Response:
Thank you very much for your comment. Yes, more than ten grass carp pituitaries were used for each experiment repeated. One grass carp pituitary could only produce about 6 million cells, the cell density in in vitro experiment was 2.5 million cells/well in 24-well plate. So we need about more than ten grass carp pituitaries for each 24-well plate. (Line 225-226)
(3) At least two independent experiments showing essentially the same results should have been done. And, in each experiment, how many replicates were done for each treatment (i.e., the mean ± SEM refers to how many analyses)? I do not understand what is meant by “the data presented (expressed as Mean ± SEM) were pooled results from 6-8 experiments” (lines 249 and 250). Could it be clarified?
Response:
Thank you very much for your comment. There are six replicates from two independent experiments for each treatment. Sorry for our short coming, “the data presented (expressed as Mean ± SEM) were pooled results from two independent experiments”. (Line 265-267)
Table 1 (letters too small, too much information) and Figure 1D (the name of the organs/tissues analyzed) have poor quality and Figure 2B and D are difficult to interpret (the lines are overlapped). Regarding also Figure 2B and D, the symbols and colors used for each peptide should be maintained in both graph because it gets easier to interpret and compare the results.
Response:
Thank you very much for your comment.
As you suggested, (i) the Table 1 was numbered as Supplemental Table 1, and the letters in the table was more clear.
(ii) The name of the organs/tissues in Fig.1D were more clear in the revised manuscript. (Line 286)
(iii) The symbols and colors used for each peptide have been maintained in Fig.2B and 2D. (Line 338)
For Figures 1D and 3A it should be indicated in the legend what is “-ve”. I guess is the negative control but it should be indicated.
Response:
Thank you very much for your comment. As you suggested, the “-ve” was replaced with “negative control” in Fig. 1D. (Line 286)
The authors use letters like a, b, c, d, e, f in Figures 2, 3, 4 and 5 to label statistical significance (instead of stars). They should provide the meaning in the figure legends; I noticed that in Figure 3, 4 and 5, the authors refer “the groups denoted by different letters represent a significant difference at p<0.05 (ANOVA followed by Dunnett’s test)” but I think it is not clear enough. It took me some time to understand, especially in Fig 2, where nothing is referred in the legend.
Response:
Thank you very much for your comments. You are right. The letters like “a, b,c, d” is confusing and not clear. As you suggested, we have replaced them with “*” or “**” in all figures (Fig. 2, Fig. 3, Fig. 4, Fig. 5).